# Hydrogen Ion Dynamics of Cancer and a New Molecular, Biochemical and Metabolic Approach to the Etiopathogenesis and Treatment of Brain Malignancies

**DOI:** 10.3390/ijms20174278

**Published:** 2019-09-01

**Authors:** Salvador Harguindey, Julian Polo Orozco, Khalid O. Alfarouk, Jesús Devesa

**Affiliations:** 1Institute of Clinical Biology and Metabolism, 01004 Vitoria, Spain; 2Al-Ghad International Colleges for Applied Medical Sciences, Al-Madinah Al-Munawarah 42316, Saudi Arabia; 3Alfarouk Biomedical Research LLC, Tampa, FL 33617, USA; 4Scientific Direction, Foltra Medical Centre, 15886 Teo, Spain

**Keywords:** etiopathogenesis of gliomas, pH-centric anticancer paradigm, pH, NHE and proton extruders, reverting proton reversal, cellular acidification in gliomas, treatment of glioblastoma multiforme

## Abstract

The treatment of cancer has been slowly but steadily progressing during the last fifty years. Some tumors with a high mortality in the past are curable nowadays. However, there is one striking exception: glioblastoma multiforme. No real breakthrough has been hitherto achieved with this tumor with ominous prognosis and very short survival. Glioblastomas, being highly glycolytic malignancies are strongly pH-dependent and driven by the sodium hydrogen exchanger 1 (NHE1) and other proton (H^+^) transporters. Therefore, this is one of those pathologies where the lessons recently learnt from the new pH-centered anticancer paradigm may soon bring a promising change to treatment. This contribution will discuss how the pH-centric molecular, biochemical and metabolic perspective may introduce some urgently needed and integral novel treatments. Such a prospective therapeutic approach for malignant brain tumors is developed here, either to be used alone or in combination with more standard therapies.

## 1. Introduction

Our interest in the treatment of malignant gliomas (MG), mainly glioblastoma multiforme (GBM), dates from more than thirty years ago [1], and interest in the relationship of pH and cancer dates even further back to a decade earlier [2]. In order to describe the new pH-centric approach to malignant diseases, the concept of “hydrogen ion dynamics of cancer” was introduced in the early 1980s [3], a term that is increasingly used in publications in this growing field in cancer research and treatment [4].

In spite of all the efforts undertaken during the last few decades in clinical neuro-oncology and the new drugs, therapeutic methodologies and protocols employed, the prognosis of brain cancer is still dismal, with GBM showing the worst five-year survival rates among all cancers [5]. Very few patients survive more than a year, and those that do usually suffer a deteriorated quality of life, particularly in brain stem glioblastomas in children and young adults [6,7]. So far, only a few isolated cases of long-term survivors have been reported following the orthodox and traditional therapeutic approaches of maximal surgical resection, chemotherapy, immunotherapy and radiotherapy [8]. 

The cause of most brain tumors is unknown. Also, there are no environmental-associated conditions that are known to cause these malignant brain tumors, with the exception of ionizing radiation. Therefore, prevention is not a possibility. Besides, the initially promising results of intra-arterial chemotherapy have shown no real benefit in the long term, especially considering the serious and even life-threatening side effects that these procedures can induce [9]. In order to improve the regrettable therapeutic situation of MG, large multicenter studies have proposed the concerted use of several drugs through combined therapeutic protocols [10]. An interesting combination of nine different repurposed drugs have been tried recently, along with standard therapies, although they have only resulted in marginal benefits [11].

## 2. Towards a New Perspective and Clinical Approach to Malignant Gliomas 

The overall therapeutic failure in the treatment of MG makes it necessary that brand new ideas and untrodden approaches and efforts are urgently developed in order to improve the therapeutic results. The different areas of the pH-centered perspective of cancer, from etiopathogenesis to therapeutics, have been recently summarized in a recent issue dedicated to this area of research and treatment. However, MG were not considered in these publications of our group, this being the first contribution in the literature dedicated to MG [12,13]. At the same time that these conceptual changes in oncology were taking place, the interest raised by the abnormalities of the intracellular and the extracellular pH (pHi and pHe) of both normal and brain cancer cells have shown a most beneficial “side effect” in the understanding the many levels of the pH-related paradigm, and/or the hydrogen ion (H^+^) dynamics of cancer [3,13,14,15,16,17,18,19,20,21].

### 2.1. Pathological Hydrogen Ion Dynamics and Acid-Base Homeostasis in the Etiopathogenesis of Brain Tumors and Other Malignant Processes: Genetic and Epigenetic Factors

#### 2.1.1. On Etiopathogenesis

Regarding the energetics of cellular acid-base homeostasis, different proton transporters (PTs) have been implicated in the etiopathogenesis of MG. Several PTs, proton pumps (PPs) (Vacuolar-ATPases), ion channels, ion exchangers and aquaporins have been reported to be involved in the pathophysiology of brain malignancies, mainly because of their capacity to induce an intracellular alkalinization (IA), which at the same time induces multidrug resistance (MDR) [12,22,23,24,25]. Specifically, an elevated pHi mediates in the MDR resistance of MG [26], but also in other malignancies, such as cervical and ovarian cancer, to different drugs like cisplatin, vinblastine and 5-fluorouracil [27]. Contrariwise, the suppression of the activity of V-ATPase potentiates the cytotoxic effects of cisplatin by decreasing pHi [28]. 

Most importantly, cisplatin can significantly modify the intracellular pH of cancer cells inducing cytoplasmatic acidification, seemingly through a cisplatin-mediated inhibition of proton extrusion and down-regulation of NHE-1. Otherwise, the activity of NHE-1 and its effect on increasing pHi also increases cisplatin resistance to treatment [29]. As a result of these findings, both NHE and the cancer-specific pHi/pHe proton reversal can nowadays be considered to be the new and main key actors in the etiopathogenesis of MG. Indeed, the main molecular, biochemical and metabolic factor blamed for the onset of MG is the NHE exchanger isoform 1 (NHE1) and, secondarily, other NHE isoforms like NHE5 and NHE9 [30]. In summary, during the last few years, the new hydrogen ion (H^+^)-related perspective of a wide array of cancers, and of MG too, has progressed from the original Warburg approach to glycolytic cancer metabolism to a post-Warburg pH-centered paradigm. In the same line, and since the 1980s, the interest in cancer etiopathogenesis has shifted its main emphasis from intermediary metabolism and glycolysis to the selective H^+^ abnormalities of cancer cells and tissues and on its molecular relationships with H^+^-extruding transport systems and pHi/pHe regulators [31,32]. 

Thus, NHE1 has become a specific and major factor in the etiopathogenesis of MG regarding its etiology, migration, survival in hostile microenvironmental conditions and relentless progression [33,34,35]. NHE 1 un-regulation maintains an intracellular alkaline pHi, which also prevents an intracellular acidification (IAc) as the main mediator of a selective apoptosis of MG cells. Thus, an elevation of pHi also represents a key factor of multiple drug resistance (MDR) in MG, similar to other malignancies [22,23,34,36,37]. 

In the same line of cancer etiopathogenesis, the extracellular and/or microenvironmental tumoral acidification (↓pHi), induced as a consequence of the initial intracellular alkalotic deviation (↑pHi), secondarily becomes a fundamental issue of a primary importance in cancer growth. In this way, this acidic pHe becomes the ultimate mechanism to allow malignant tumors to escape from the anti-tumor immunity of the parasitized human organism. The final result is that this microenvironmental-intratumoral-extracellular (EC) low pHe situation creates a protective shield around cancer with the onset of a state of energy and immunosuppression mediated by such EC acidification-induced losses of function of T and NK cells [38,39,40,41,42].

It is worth recognizing that even in the case of a genetically-induced overexpression of NHE1 [43], genes do not seem to exert a direct influence on cellular metabolism, but they do it through the microenvironmental changes they induce. In this vein, F. Nijhout [44] integrated both ways, genetic and epigenetic, by stating: “When a compound modified by a gene is needed, it is a signal from the environment which activates the expression of the gene and never an intrinsic characteristic of the gene”. From all the late results in the field it can be concluded that NHE1 also plays a fundamental role in the local growth and activation of the metastatic process of many other malignant tumors besides MG [22,35,37,45,46,47,48]. Most significantly, the overexpression of NHE1 can be considered a widespread carcinogenic factor that is stimulated by myriad elements of different natures, all of which induce a high pHi-mediated carcinogenic response in normal cells of many different origins and irrespective of their genetic background and location (Table 1). It is most likely that at least some of these contributing factors to the etiopathogenesis of cancer can also be involved in the pathogenesis of MG through their effects on overexpressing and/or up-regulating NHE1 and the intracellular alkalinization induced by it.

We are not aware that the possibility of a cause-effect relationship of genetic mutations of *BRCA1* and *BRCA2* with pHi and/or NHE1 expression has been made before. The intention of including *BRCA1* and *BRCA2* in Table 1 is to suggest that NHE1 and/other proton extruders, like carbonic anhydrases (CAs), can mediate in the carcinogenic action of these gene mutations in a similar, or even the same, way as happens with other gene products [44]. Table 1 also shows the many hormones, growth and trophic factors, as well as certain cytokines that over-express NHE1 and induce its pH-related pathological effects on cellular metabolism [13,21]. Human growth hormone (HGH) on its own is able to stimulate the production of a wide array of growth factors, hormones and cytokines, such as IGF-1, EGF and its receptor, VEGF, FGF, EPO, BDNF, PDGF, certain interleukins and sex steroids, some of which up-regulate NHE1 [50]. (Also see Figure 1).

During the life of Otto Warburg (Warburg died in 1970), proton extruders and all the other factors shown in Table 1 were not known. Thus, Warburg could not know that cancer cells were not acidic, as he always thought, but just the opposite [51,52]. Furthermore, we also know that Warburg was wrong in defending the theory that aerobic glycolysis was the prime cause of cancer, but it can also be said that the prime cause of cancer is the main, and perhaps universal, mediating cause of aerobic glycolysis, namely, intracellular alkalinization [13]. Indeed, this and other recent publications have also led to the conclusion that the famous Warburg Effect may be completely explained through the elevation of pHi in cancer cells [53,54,55]. 

NHE1- and pH-related pathology are receiving increased levels of attention as fundamental factors in other areas of carcinogenesis. In this vein, Hardonnière et al. have advanced a most provocative and integral explanation of human environmental carcinogenesis [56]. Even more recently, the same research group has correctly suggested that the oncogenic activity of many carcinogens of different origins and natures (Table 1) can share the same and/or similar pathways and effects on cellular H^+^ dynamics, facilitating proton gradient reversal. This opens up the possibility that the overexpression of either NHE1 and/or other proton extruders could be behind the existence of a universal mechanism responsible for the induction of environmental carcinogenesis [4]. 

The findings of these researchers, but also of other groups, suggest that the final cancer-inducing mechanisms of carcinogens like polycyclic aromatic hydrocarbons, as well as the activity of other widespread carcinogenic environmental compounds ubiquitously present in low concentrations in nature, even in groundwaters, like arsenic salts, are NHE-mediated [57]. These findings lead towards a unitarian synthesis of environmental carcinogenesis and to the conclusion that there could well exist a final and universal mediating acid-base mechanism, namely, a mechanism related to environmental H^+^ dynamics that can fully explain human carcinogenesis. Finally, Figure 1 graphically shows the growth and trophic factors whose effects are mediated by the NHE1 antiporter as H^+^ extruder, some of them also stimulating tumoral angiogenesis. For a more complete review of NHE-related proangiogenic and antiangiogenic molecules, see ref. [58].

Although in Figure 1 the human growth hormone (GH) appears as a pro-oncogenic factor, it has also been shown than this hormone can be a useful and safe treatment for many different pathologies, even in some neurodegenerative processes [13,21,59,60]. However, even in patients with a past history of neoplasia, GH replacement therapy does not appear to increase the chances of inducing a tumoral process [61,62]. Contrariwise, GH has also been considered as a “one-step” oncogene able to promote both proliferative and metastatic processes [63,64,65]. In this vein, since the existence of GH receptors in MG has been demonstrated, it seems that GH could exert actions responsible for the induction, or at least progression, of MG [66]. It can then be assumed that although systemic GH may lack a direct effect on the induction and/or progression of these tumors, some growth factors induced by GH, like IGF-1, EGF and VEGF, could negatively affect tumor growth through NHE stimulation and/or cellular alkaline pH changes, an effect that was first described from seminal publications decades ago (Figure 1) [67,68,69].

#### 2.1.2. On Treatment 

From the opposite point of view of cancer etiopathogenesis, namely, therapeutics, the pharmacological targeting and inhibition of NHE1 and other ion transporters, pumps and voltage gated sodium channels, is fundamental in inhibiting both local growth and the different stages of the metastatic process, either in MG and/or in a variety of other extracranial malignant tumors [70,71,72,73,74,75,76,77]. 

The H^+^-related perspective, as applied to the treatment of MG, has been defended by different research groups [49,78,79,80,81]. In this line, inhibiting NHE1 in MG acidifies tumor cells while normal astrocytes are not affected, a finding that open the way towards a selective and non-toxic, or minimally toxic, treatment of MG [35]. In a similar context, this therapeutic approach has been most correctly called “the Achilles heel of cancer” [82], although it also appears not to be free from some therapeutic limitations [83]. 

Furthermore, blocking Na^+^/H^+^ exchange decreases tumor growth and stimulates MG immunogenicity, besides increasing the effect of temozolomide (TMZ) [34,84]. Most surprisingly, these authors also reported that TMZ increases NHE1 protein levels in human glioblastoma cells, a feature that not only could increase TMZ resistance, but raises serious doubts about a possible deleterious effect of TMZ in the treatment of MG. On the other hand, the combination of TMZ and cariporide in a mouse glioma xenograft model significantly prolonged the survival of the mice in the same report. Cariporide (HOE642) has been used in human trials but only in a cardiological setting, and although it has been repeatedly proposed as an anticancer drug in either brain cancer and/or in many other malignant tumors because of its effect as a selective intracellular acidifier of cancer cells of many different lineages, it has never reached any clinical or even preclinical trials in human oncology [37,49,85,86].

Other membrane-bound ion transporters and related mechanisms besides NHE overexpression/inhibition are of fundamental importance in the acid-base regulation of MG, mainly lactate and pH-related glucose transporters and glycolysis [55], MCTs (monocarboxylate transporters), mainly MCT1 and MCT4, and carbonic anhydrase IX (CAIX) [87]. For an integral review on this subject, see ref. [78]. Furthermore, targeting MCTs and CAIX concomitantly with NHE1 inhibition offers a highly promising and integrated approach to the treatment of MG [88,89]. These concepts are the scientific foundation of the treatments proposed in Table 1.

Inhibiting lactate extrusion in MG has also been hypothesized to increase sensitization to radiotherapy by delivering small-molecule MCTs inhibitors to the tumor bed or to the postsurgical resection area [90]. MCT has also been reported to inhibit the invasiveness of GBM while inducing necrosis under different circumstances [70]. Last but not least, a drug like cisplatin, used in MG and many other tumors, has been found to induce IA in vitro and in vivo, a feature that explains whether or not an individual patient responds to treatment with this drug, and perhaps to many others as well [28]. 

Some ion channels are significantly involved in both the regulation of pHi/pHe in cancer cells and in the acquisition of their proliferative and pro-invasive capacities. Ion channels regulate several cell processes, such as cell proliferation, resistance to apoptosis, cell adhesion, cancer cell motility and extracellular matrix invasion. Consequently, an altered physiology of ion channels has also been proposed as a new hallmark of cancer cells and as a potential target for selective therapeutics, either in glioma or in other malignancies [91,92,93,94,95,96,97].

Persistent NHE1 activity in glioma cells is also consistent with depolarized membrane potentials, calcium loading, high pHi and increased cellular Na^+^ levels. While inhibition of NHE1 by cariporide alone seems not to be toxic to glioma cells, its combination with the inhibition of the Na^+^/Ca^2+^ exchanger NCX1.1 selectively kills brain tumor cells [98]. This is consistent with the growing evidence that Ca^2+^ homeostasis is importantly remodeled through the involvement of multiple Ca^2+^ channels and transporters [97]. These molecular mechanisms, that take place at the plasma membrane or in intracellular compartments, participate in enhanced proliferation, cancer cell survival and invasion. As a retro-feedback, intracellular Ca^2+^ concentrations also modulate the activity of NHE1 [99]. Finally, the inhibition of voltage-gated sodium channels, either on their own or through interaction with NHE1, has been proposed in order to prevent cancer growth and progression as well as the metastatic process in different experimental conditions and contexts [13,100].

### 2.2. General Principles of Low pHi-Dependent Cancer Cell Apoptosis as Applied to the Clinical Treatment of Malignant Gliomas 

From the point of view of metabolism, the therapeutic strategy to be recommended in human cancer is mainly directed to achieve an apoptosis-inducing IA of malignant cells while sparing normal cells. This IA must achieve a pHi low enough to induce a chain reaction that leads to selective apoptosis [13,21,43,101,102]. During this apoptotic process the endonuclease DNAase I becomes another fundamental mediating factor [103]. Similarly, a low pHi-induced apoptosis is also mediated through other different mechanisms [21], while the Bcl-2 anti-apoptotic protein seems unable to inhibit it [104]. This is why IA is the main and fundamental weapon of the pH-centered anticancer treatment [102]. IA can be achieved through inhibition of NHE1 and/or H^+^-ATPase and/or CAs IX and XII and/or successful chemotherapy. Most recently, a full issue has been published on the importance of CAs in cancer and other non-oncological pathologies [76].

In all these cases, cells will go into an acid-mediated metabolic collapse and catastrophe which is followed by apoptosis or necrosis [32,105,106]. Other mechanisms of successful cancer treatment include overcoming blockades to cancer cell apoptosis by inhibiting the Bcl-2 anti-apoptotic family, whose activity is, not surprisingly, mediated by intracellular (ic) alkalinization (IA) and inhibited by intracellular (ic)acidification (IAc) [104,107,108,109,110,111,112]. Furthermore, it has been known for a number of years that superoxide (SO) formation, as induced by hormonal or growth factor deprivation, also induces apoptosis of brain cells [113,114], a phenomenon that is further increased after reacting SO with nitric oxide (NO) to form peroxynitrite [115,116]. In contrast, low or physiological concentrations of NO prevent apoptosis [117]. In summary, the different killing mechanisms in cancer cells all share one single and key feature: that they induce a degree of ic acidification incompatible with cellular life. This also contributes to reverse the cancer-specific and pathological hydrogen ion (H^+^) abnormal dynamics of cancer cells and tissues.

#### 2.2.1. An Integrated Approach to Treatment

In spite of all the intratumoral and microenvironmental odds, GBM cells manage to maintain a normal to alkaline intracellular acid-base status in order to protect themselves from a threatening pro-apoptotic IAc. This represents an important part of what we initially called “the neostrategy of cancer cells and tissues” (Figure 2) [13,22,118,119].

IA may be useful as a diagnostic tool, since the potent NHE inhibitor cariporide (HOE642) helps to localize GBM by inducing IA of brain tumors in mice [86]. In the same line, the determination of the effects of different anticancer drugs on pHi is a new and most promising area of research, not only in therapeutics but also in the diagnosis and degree of extension of MG, at least in GBM. This new pH-centered tool allows us to test, even in individual tumors, the sensitivity to IA of different pH-sensitive anticancer drugs such as topiramate, lonidamide, quercetin, dichloroacetate (DCA) and cariporide [30,36,120,121,122,123,124,125,126]. Finally, increasingly sophisticated methodologies to determine pHi and pHe in malignant tumors are being published, while recent and complete reviews on the subject are also available [127].

#### 2.2.2. Intracellular Acidifiers and Anti pH-Related Drugs with Potential Activity in the Treatment of Malignant Gliomas

One of the reasons why most MG are currently incurable is the very high and rapid invasiveness they show, at least in the case of GBM. Their extracellular acidity (EA) is a basic requirement for their highly invasive capacity [127]. Importantly, a two-edged or even triple-edged treatment that addresses simultaneously glucose availability and the pH alterations found in brain tumors may represent an important step forward towards decreasing the migration/invasion process of GBM [102].

Table 2 shows the intracellular acidifiers that through different pHi-lowering mechanisms, mostly related to inhibitory effects on one or more proton extruders (NHE1, CAs, MCTs, H^+^-vacuolar ATPases, etc.), or as voltage gated sodium channels (VGSC) inhibitors, have shown activity against gliomas and promise a potential benefit in the treatment of MG and/or GBM. However, it is most likely that the utilization of only one of them would not be sufficient to regress or even control brain cancer. Thus, a concerted utilization of several of these drugs in pharmacological doses, as has been repeatedly suggested, is mandatory [48,100,128,129]. This should be accompanied by the implementation of cooperative methods that, at least in the case of some of these new and/or repurposed drugs, can overcome the pharmacokinetic blockade imposed by the blood–brain barrier (BBB). To achieve this, the use of polymers, liposomes and nanoparticles has been proposed as a useful method to increase drug delivery to the brain in spite of the BBB [130,131]. In the same line, and in order to increase the efficacy of targeted drug delivery systems with low toxicity, acridine orange, a photodynamic-related acidophilic dye with a strong tumoricidal action in different malignancies, takes advantage of the low extracellular pH of tumors [132,133]. Other photodynamic-oriented procedures have been recently reported as selective and local intracellular acidifiers in cancer treatment [134].

Finally, for a more complete exposure of pH-related acidifiers as anticancer and/or repurposed drugs in preclinical and clinical oncology, see [13,128].

## 3. Conclusions

Glioblastoma multiforme and other malignant brain tumors have a very poor prognosis that shortens both human life and severely worsens its quality in patients with these diseases. The results of standard treatments have been dismal, with minimal and non-significant improvements over recent decades. The treatment of glioblastoma multiforme is, perhaps, the major failure of modern oncological practice. Thus, a different approach is urgently needed to treat this disease. This contribution tries to offer a different approach to that of the orthodox treatments of conventional and mainstream oncological practices in order to implement new therapeutic combinations through schemes that are aimed to reverse the cancer-selective pH-related abnormalities of cancer cells and tissues. For the first time, an integrated and advanced model is considered as a rational approach and alternative proposal for the treatment of malignant brain tumors. Furthermore, the main aim of this pH-centric perspective is to break through the therapeutic impasse and skepticism that dominates this important and disappointing area of modern oncology. The measures proposed here should be tried as an integrated scheme, since its partial utilization would not achieve an efficient glioma cell hyperacidification and selective apoptosis of brain malignancies. This acid-base change, key to the selective energetics of the biochemistry and metabolism of all cancer cells and tissues appears as the most promising target in the treatment of brain cancer. Its full benefits still require further translational research and the activation of clinical trials in bedside neurooncology. We hope that this original perspective may make a difference in the treatment of malignant gliomas on a short-term basis.

## Figures and Tables

**Figure 1 ijms-20-04278-f001:**
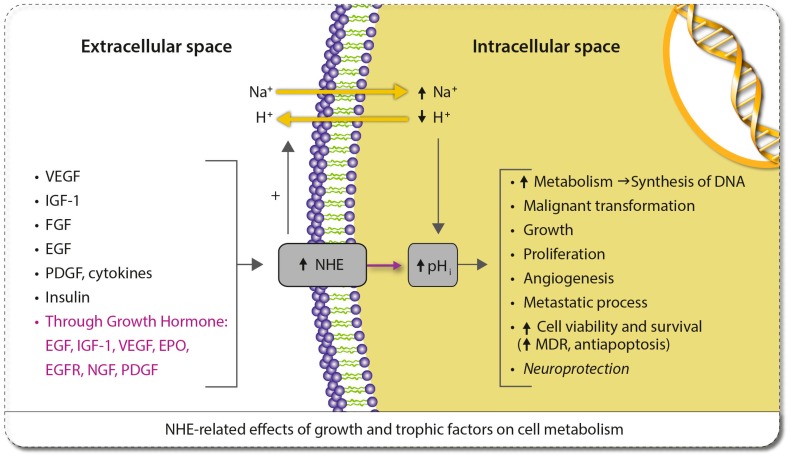
Growth and trophic factors, and cytokines that are involved in the carcinogenic expression and/or hyperactivity of NHE1 and the consequent increase in pH (modified and updated from ref [13]).

**Figure 2 ijms-20-04278-f002:**
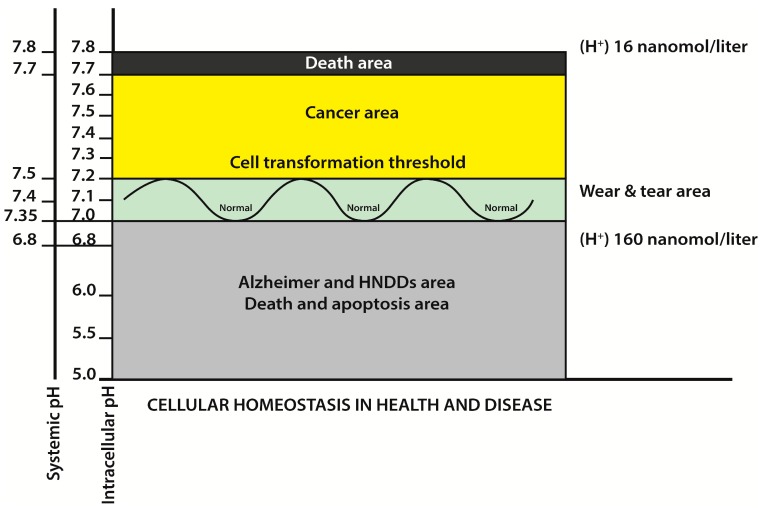
The main aim of the pH-centered treatment is to induce a selective low pHi-mediated apoptosis of all cancer cells. This figure also shows the opposite pH ranges in cancer and in human neurodegenerative diseases (HNNDs). The higher the cellular pH, the lower the hydrogen ion concentrations, and vice versa. For further details, see [13,21].

**Table 1 ijms-20-04278-t001:** This Table (modified and updated from ref [49]) shows some carcinogenic factors that increase cellular pH through up-regulation of NHE activity.

Carcinogenic Factors That Increase Cellular pH Through Up-Regulation of NHE Activity
Proton transporters-extruders (PTs) and proton pumps (PPs)
Virus (HPV E5 virus: human papiloma virus)
Oncogenes and viral proteins (v-mos, Ha-Ras, HPV16 E7))
Gene products (Bcl-2)
p53 deficiency
Genetic instability and mutations (BRCA1 and BRCA2?)
Chemical carcinogens (benzo(a)pyrene, polycyclic aromatic hydrocarbons, arsenic salts in groundwaters
Chronic hypoxia and HIF
Different mitogens
Hormones and cytokines (Insulin, Growth Hormone, Prolactin, Glucocorticoids, IGF-1, EGF, VEGF, PDGF, Il-1, Il-8, GCSF, TGFß, Angiotensin II, PGE2, Bombesin, Diferric transferrin
Glucose overload
Ageing (“Time causes cancer”- Otto Warburg)

**Table 2 ijms-20-04278-t002:** Drugs with a potential benefit in the treatment of malignant brain tumors that, to a large extent, have not yet been clinically or even preclinically tested.

Drug	Dose and Side Effects	Objective
Topiramate	Starting dose, 50 mg twice a day. The dose must be increased 50 mg every week until reaching 200 mg twice a day.	Topiramate is a voltage gated sodium channel inhibitor that acidifies glioma cells and reduces the risk of seizures [120].
Acetazolamide (AZM)	Starting dose, 125 mg twice a day the first week. And 250 mg twice a day after the first week.	Acetazolamide is a carbonic anhydrase (CA) pan-inhibitor and cell acidifier [76,77].
Amiloride (and/ or liposomal amiloride)	10–30 mg three times a day. Hyperkaliemia can be an occasional problem, more with non-liposomal amiloride.	Amiloride is a non-specific NHE inhibitor and the first one that was developed and introduced in the clinic as a K^+^ sparing diuretic [135]. A positive clinical experience in an occasional patient has been reported [136,137]. Non-liposomal amiloride barely crosses the blood–brain barrier (BBB).
Quercetin	There is no established dose for quercetin. Oral doses of 3 g three times a day are well tolerated in the long term. Very poor oral absorption.	Quercetin is a flavonoid sold over the counter as a nutraceutical, a pan-monocarboxylate transporter (MCT) inhibitor and intracelllular MG acidifier [123,125,138]. Liposomal quercetin is also available.
Fenofibrate	100 mg twice a day.	Fenofibrate is a PPRα agonist that reduces the motility of glioma cells [139], induces their apoptosis [140,141,142,143,144], inhibits glycolytic metabolism [145] and reduces migration [139,146,147]. Fenofibrate also targets glioma stem cells [148,149]. For a further review on fenofibrate see [150], and for a review on fenofibrate in glioma, see [151].
Celecoxib	400 mg twice a day.	Celecoxib inhibits growth and induces apoptosis [152,153,154,155,156,157,158,159,160,161,162]. It also increases the effectiveness of chemotherapeutic drugs [163,164,165,166,167,168,169,170,171] and radiotherapy [172,173,174,175,176]. It attenuates de Wnt/βcatenin pathway [177], reduces angiogenesis [178,179,180,181] and inhibits myeloid derived suppressor cells [182]. For a review on celecoxib in glioma, see [183] and [184].
Cariporide (HOE642) (Unavailable in oncology)		Cariporide (HOE 642) is a powerful NHE1 inhibitor but, unfortunately, is not available for clinical use in oncology. It is orally bioavailable [49]. It also induces non-apoptotic cell death in malignant glioma [98].
Diclofenac (Usual doses)		Diclofenac inhibits lactate formation and counteracts immune suppression in a murine glioma [75].
Dichloroacetate (DCA)	25–40 mg/kg daily in 2–3 weeks cycles (plus Vitamin B1).	DCA is orally available and has been used frequently for GBM in the experimental context as a cell acidifier and glycolytic inhibitor [185,186,187,188], as well as in phase I clinical trials [189,190,191].
Betulinic acid (clinical trials are underway)	Different dosages.	It penetrates the BBB and is highly effective in temozolomide-resistant glioblastoma cells [192,193]. It is also effective against other tumors, like melanoma and neuroectodermic tumors. Its antitumoral activity is also related to its effect as a topoisomerase I inhibitor [194].
Cisplatin (CDDP) (Usual dosages)		Cisplatin induces pHi acidification and a metabolic shift from glycolysis to oxidative metabolism in cervical cancer cells. This is accompanied by the inhibition of cancer cell growth. Cells either recover, maintaining an alkaline pHi to survive and proliferate, although at reduced growth rates, or undergo cell death [28]. CDDP also induces a therapeutic intracellular acidification [29,30,126].
Compound 9t (C9t) (Unavailable)		C9t has been reported to be 500-fold more potent against NHE1 than cariporide and to have a greater selectivity for NHE1 over NHE2 (1400-fold). Besides, C9t is orally bioavailable, has low side-effects in mice and shows a significantly improved safety profile over other NHE1 inhibitors [195].

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
