# Peer review of "Hydrogen Ion Dynamics of Cancer and a New Molecular, Biochemical and Metabolic Approach to the Etiopathogenesis and Treatment of Brain Malignancies"

_ijms, 2019, doi:10.3390/ijms20174278_

Round 1

Reviewer 1 Report

In this review, authors demonstrated that the cellular acidity of cancer cells can be considered as a new target for cancer therapy. Even though the concept is interesting, the way how described the concept in this manuscript is similar to the authors’ previous review article (Seminars in Cancer Biology 2017, Volume 43, Pages 157-179). I cannot find clear differences between this manuscript and the authors’ previous paper except the table for the intracellular acidifiers. To publish this manuscript, authors should revise the manuscript by minimizing the redundant information and focusing on the new/recent research’s trends such as clinical applications based on the low pHi-induced apoptosis.

Figure 1 was adapted and updated from the authors’ previous review article (Seminars in Cancer Biology 2017, Volume 43, Pages 157-179). Please cite the reference in the figure legend.

Author Response

Comments and criticisms of Reviewer No. 1:

Referee comment:

Main comments of the referee report: (3 out of 5):

The work is well organized and comprehensively described. The work is scientifically sound and not misleading The references are adequate and relate to previous work (x) English language and style are fine/minor spell check required

Other comments by this referee                                                                                                           a) In this review, authors demonstrated that the cellular acidity of cancer cells can be considered as a new target for cancer therapy. Even though the concept is interesting, the way how described the concept in this manuscript is similar to the authors’ previous review article (Seminars in Cancer Biology 2017, Volume 43, Pages 157-179). I cannot find clear differences between this manuscript and the authors’ previous paper except the table for the intracellular acidifiers. To publish this manuscript, authors should revise the manuscript by minimizing the redundant information and focusing on the new/recent research’s trends such as clinical applications based on the low pHi-induced apoptosis.

Reply by authors:

Originality of the work. So far, only one short and recent publication coming from an Iranian research group has addressed the main subject of our contribution on pH and malignant gliomas. (See Reference No. 29 of our list of references by Tamtaji, O.R.; Mirzaei, H.; Shamshirian, A.; Shamshirian, D.; Behnam, M.; Asemi, Z., New trends in glioma cancer therapy: Targeting Na(+) /H(+) exchangers. Journal of cellular physiology 2019, 1-8. DOI: 10.1002/jcp.29014.

Since traditional and standard oncology and neurooncology are still largely unaware of even the existence of the new pH-centric anticancer paradigm, we have dedicated around a 15% of this contribution to briefly introduce the new pH-related concepts to the reader. The rest, around 85% of this work on brain malignancies, is entirely new in the oncological literature - apart from the Iranian paper already mentioned - and has not even mentioned in any other previous reviews like the one that this referee mentions (S. Harguindey, et al. Seminars in Cancer Biology 2017, Volume 43, Pages 157-179).

In any case, the entire text now resubmitted has been thoroughly improved, misspellings corrected, sentences shortned, etc.,  and any possible redundancies we have been able to identify have been attenuated or suppressed to the best of our ability.

Also, Figure 1 has been reformatted as a Table, and the latest information not present in any other previous publication, either of our group or others, inserted in it.

The other table and the two figures are entirely original for this contribution. All in all, we have to disagree with the referee statement where he mentions:  “I cannot find clear differences between this manuscript and the authors’ previous paper except the table for the intracellular acidifiers”, since both manuscripts are entirely different no matter both of them deal with the same molecular, biochemical and metabolic perspective, although this time specifically and selectively applied to brain cancer.

b) Referee comment: Figure 1 was adapted and updated from the authors’ previous review article (Seminars in Cancer Biology 2017, Volume 43, Pages 157-179). Please cite the reference in the figure legend.

Reply by authors:

First, we believe that the reviewer refers to Table 1, not to Figure 1.                                                         

The reference that this referee suggests to introduce in the legend is No. 13 in the list of references, and this is considered further down in the legend. There the legend reads:  “This table also shows the many hormones, growth and trophic factors, as well as certain cytokines, that over-express NHE1 and induce its pH-related pathological effects on cellular metabolism [13,21].                                                                                                                                           Reference  No. 13 is the original publication where a less developed Table was published by our group, so the reference to it in the figure legend was already mentioned.

Reviewer 2 Report

This review is timely and very comprehensive. Overall, it is well written and the topic covers an area of importance where there is a current dearth of information about pH regulation in gliomas, and how this may contribute to cancer progression. The literature has put into perspective both past and present research in the field of proton transport and, indeed, ion transport as a carcinogenic signal. It also discusses the metabolic changes that occur in cancer cells in the perspective of this intriguing pH-centric paradigm to treating cancer.

Recommend for publication with minor revisions:

Table 1 was not in a table format in the text copy received so it could not be properly evaluated. Please check formatting on this.

In Table 2, cariporide is discussed as not currently being available for clinical use. It might strengthen the review and provide some context to briefly discuss (and/or reference) the outcome of cariporide use in past clinical trials though these were not done in cancer patients.

Remove the use of colloquialisms e.g. “nowadays” (lines 50, 81, 199).

Grammatical errors:

Line 58: “here considered” should be “considered here”

Line 66: “80’s” should be “80s”

Line 73: “reported a few isolated cases” should be “a few isolated cases reported”

Line 105: “processes. Genetic” should be “processes: Genetic”

Line 128-132: Very long sentence that is confusing: e.g. “...tumors NHE1, genetically and/or induced....mechanisms, NHE1 is a specific...”

Line 135: “buta also” should be “but also”

Line 145: “It is” should be “It is”

Line 201: “glycolisis” should be “glycolysis”

Line 223: Remove “cell” before extruder.

Line 236 and 245: “In spite that” should be “though”

Line 301: “implication” – Is this the correct word choice? Confusing.

Line 302: “mechanisms are expressed” should be “mechanisms occur”

Line 327: should be “anti-apoptotic”

Line 335-337: Confusing; used “reversal” multiple times. Reword.

In Figure 2: Alzheimer “y” HNDDs area. Is “y” meant to be “and”?    

Line 355-360: Very long sentence. Confusing at “...apart that it allows to test...” Consider rewording.

Line 369: “o”?

Line 375: “voltage gate” should be “voltage gated”

In Table 2: Topiramate under Objective: missing “inhibitor” after VGSC.    

Line 415-416: “here advanced” should be “proposed here”

Author Response

Reviewer No 2 comment:  Remove the use of colloquialisms e.g. “nowadays” (lines 50, 81, 199).

Reply by authors:                                                                                                                                            These colloquialisms in lines 81 and 199 have been have been removed.                  Line 50, says: “Some tumors with a high mortality in the past are curable nowadays. However, there is one…”.                                                                                                                        Sorry, but we do not appreciate any colloquialism in line 50, unless the referee considers that it would be better to remove the word “However,”                                                                                           

So, we have not change it.

Reviewer: Grammatical errors and authors corrections:

 Line 58: “here considered” should be “considered here”.                                                                   

This has been corrected. 

Line 66: “80’s” should be “80s”                                                                                                         

This has been corrected. 

 Line 73: “reported a few isolated cases” should be “a few isolated cases reported”                               

This has been corrected. 

 Line 105: “processes. Genetic” should be “processes: Genetic”              

This has been corrected. 

 Line 128-132: Very long sentence that is confusing: e.g. “...tumors NHE1, genetically and/or induced.... mechanisms, NHE1 is a specific...”                     The sentence has been shortened and corrected as sufggested by the referee.

 Before corrections: As it happens in other malignant tumors NHE1, genetically and/or induced overexpressed, and either mediated by autocrine or paracrine mechanisms, NHE1 is a specific and major player in the etiopathogenesis of gliomas regarding etiology, migration, survival and relentless progression, as there is a significant up-regulation and expression of NHE1 in glioma cells [32-34].

AFTER CORRECTIONS: (Now lines 128-134). As it happens in other malignant tumors, NHE1 is a specific and major player in the etiopathogenesis of gliomas regarding etiology, migration, survival and relentless progression. This appears to be secondary to the fact that there is a significant up-regulation and expression of NHE1 in glioma cells [32-34]. NHE 1 maintains an alkaline pHi of these cells, which also prevents intracellular acidification (IA) and subsequent apoptosis of these cells.  Thus, a high pHi also represents a fundamental factor in drug resistance not only in gliomas but also in other malignancies [22,23,33,35,36].

Line 135: “buta also” should be “but also”                                                                                                    This has been corrected. 

Line 145: “Itis” should be “It is”                                                           

This has been corrected. 

Line 201 (Now line 200): “glycolisis” should be “glycolysis                            This has been corrected. 

Line 223 Now line 221): Remove “cell” before extruder.

This has been done. 

Line 236 and 245: (Now 234 and 243).  “In spite that” should be “though”

These changes have been done. 

Line 301(Now line 299):  “implication” – Is this the correct word choice? Confusing.

The word “implication” has been changed to “involvement”

 Line 302 (Now line 300): “mechanisms are expressed” should be “mechanisms occur”                                        

This change has been done by changing mechanisms are expressed” to “mechanisms take place”

Line 327: should be “anti-apoptotic”

This has been done. 

Line 335-337: Confusing; used “reversal” multiple times. Reword.

This sentence  has  been reworded, now it reads: “This also contributes to reverse the cancer-specific and pathological hydrogen ion (H+) abnormal dynamics of cancer cells and tissues.

In Figure 2: Alzheimer “y” HNDDs area. Is “y” meant to be “and”?      Yes, the Figure is corrected. Thank you.

 Line 355-360: Very long sentence. Confusing at “...apart that it allows to test...” Consider rewording.

Correction done. Now it reads: All in all, the determination of the effects of different anticancer drugs on pHi is a new and most promising area of research, not only in the treatment but also in the diagnosis and degree of extension of MG and GBM. This new pH-centered tool also allows to test, even in individual tumors, the sensitivity to IA of different pH-sensitive anticancer drugs such as topiramate, lonidamide, quercetin, dichloroacetate (DCA) and cariporide [35,120-125].

 Line 369: “o”                                                                                                                                                            “o” is now “or”                                                                                                                      Corrected

 Line 375: “voltage gate” should be “voltage gated”                                                                      Corrected                                                                 

 In Table 2: Topiramate under Objective: missing “inhibitor” after VGSC.                                  

Corrected. The word “inhibitor” has been inserted after VGSC

 Line 415-416: “here advanced” should be “proposed here

Correction done. Now it reads: “The measures proposed here should be tried as an integrated scheme since…”

Coments and criticisms by Reviewer No. 2

General comments (3 to 4 out of 5):

The work is a significant contribution to the field (punctuation 4). The work is well organized and comprehensively described. The work is scientifically sound and not measliding The references are appropriate and adequate. The English used id correct and readable.

Level of interest: An article of outstanding merit and interest in its field.

General opinion of referee:  This review is timely and very comprehensive. Overall, it is well written and the topic covers an area of importance where there is a current dearth of information about pH regulation in gliomas, and how this may contribute to cancer progression. The literature has put into perspective both past and present research in the field of proton transport and, indeed, ion transport as a carcinogenic signal. It also discusses the metabolic changes that occur in cancer cells in the perspective of this intriguing pH-centric paradigm to treating cancer.                                                                                                                   

Recommend for publication with minor revisions.

Main comments of the report: This is an excellent review, addressing an important and timely issue. It is well reasoned and has the potential to bring in a new era of research in a clinically important area. Parts of it read beautifully and are attractive to the reader. I believe the review reflects the authors’ opinions but they are well backed up by numerous references in the literature. I have only minor corrections to suggest as the manuscript does require quite a few minor typographical corrections, and a few minor conceptual ones. Overall, this is a manuscript of great interest and should be published with a high priority.

Recommend for publication with minor revisions:

Reviewer comments: Table 1 was not in a table format in the text copy received so it could not be properly evaluated. Please check formatting on this.

 In Table 2, cariporide is discussed as not currently being available for clinical use. It might strengthen the review and provide some context to briefly discuss (and/or reference) the outcome of cariporide use in past clinical trials though these were not done in cancer patients.

Reply by authors:

The entire area of the use of cariporide in previous clinical trials in cardiology was thoroughly discussed in our previous review by our group in 2017. (Reference No. 13), namely: Harguindey, S.; Stanciu, D.; Devesa, J.; Alfarouk, K.; Cardone, R.A.; Polo Orozco, J.D.; Devesa, P.; Rauch, C.; Orive, G.; Anitua, E., et al., Cellular acidification as a new approach to cancer treatment and to the understanding and therapeutics of neurodegenerative diseases. Semin Cancer Biol 2017, 43, 157-179. Since Reviewer No. 1 mentions exactly the opposite; this is, that the present text is redundant at times with that review, and the subject has been dealt with many times in recent publicationos of ours and other groups, we feel that to insist on this subject is unnecessary and would be truly redundant. Also, another review of ours is entirely dedicated to that subject too (Reference No 48: Harguindey, S.; Arranz, J.L.; Polo Orozco, J.D.; Rauch, C.; Fais, S.; Cardone, R.A.; Reshkin, S.J., Cariporide and other new and powerful NHE1 inhibitors as potentially selective anticancer drugs--an integral molecular/biochemical/ metabolic/clinical approach after one hundred years of cancer research. Journal of translational medicine 2013, 11, 282). To avoid redundancy, this reference is considered in Table 2, in the section dedicated to Cariporide. (see the original text in Table 2, in  OBJECTIVES and CARIPORIDE…)

                                                                             Cariporide (HOE 642) is a powerful                                                                                                                          NHE1 inhibitor but, unfortunately, is not                                                                                                 available for clinical useIt is orally                                                                                                                         bioavailable  [48].   …etc.

--However, in lines 267-271, (in page No. 7), the following sentence has been introduced in order to directly refer to the clinical trials with cariporide in the cardiological setting.

“Cariporide (HOE642), has been used in human trials but only in a cardiological setting, and although it has been repeatedly proposed as an anticancer drug in either brain cancer and/or in many other malignant tumors because of its effect as a selective intracellular acidifier of cancer cells of many different lineages, has never reached any clinical or even preclinical trial in human oncology [36,48,84,85].”

Reviewer 3 Report

this paper is intolerably over wordy. it is example of antiquated scientific discourse of the 1960s. examples of empty speech abound. that must be corrected.
also the authors go back and forth referring to “malignant glioma” then to gb, then to “brain tumors”. i suggest they use gb only thru out the paper. i stopped editing/reviewing at red highlighted area. tho largely grammatically correct, paper must be resubmitted after language use editing.

Author Response

Answers to Reviewer N. 3:

Comments and Suggestions from referee: this paper is intolerably over wordy. It is example of antiquated scientific discourse of the 1960s. Examples of empty speech abound. that must be corrected. Also the authors go back and forth referring to “malignant glioma” then to gb, then to “brain tumors”. i suggest they use gb only thru out the paper. i stopped editing/reviewing at red highlighted area. tho largely grammatically correct, paper must be resubmitted after language use editing.

Reply by authors to referee No. 3. (New comments to referee No. 3).

As suggested by referee No. 3, we have mainly used the term “malignant glioma” in the new MS, and only used the terms of glioblastoma multiforme and brain cancer or brain malignan- cies when strictly necessary.

This referee mentions that parts of our discourse is antiquated and belongs to the 60´s. We think that he must be referring to our publications (references 1-3) on the treatment of brain malignancies dated more than 30 years ago. This is briefly done at the beginning of the text (Background section,) where we were obliged to refer to our seminal work on the treatment of malignant gliomas and on the new pH-anticancer paradigm, a field we helped to lift and takeoff from the ground from its very beginning, and even organized the first kick off meeting on pH andCancer Metabolism in Madrid, Spain, in 2008, and thanks to the Areces Foundation.

See the next (antiquated?) text:

Background

Our interest in the treatment of malignant gliomas (MG), mainly glioblastoma multiforme (GBM), dates from more than thirty years [1], and on the relationships of pH and cancer, even dates from a decade earlier [2]. In order to describe the new pH-centric approach to malignant diseases, the concept of “hydrogen ion dynamics of cancer” was introduced in the early 80s [3], a term that is increasingly used in publications in this growing field in cancer research and treatment [4].                                                              

Calvo, F.A.; Pastor, M.A.; Dy, C.; Alegria, E.; Anton Aparicio, L.M.; Gil, A.; Harguindey, S.; Zubieta, J.L.; Martinez Lage, M., Intra-arterial and intravenous chemotherapy for the treatment of malignant glioma. Preliminary results. American journal of clinical oncology 1985, 8, 200-209. Harguindey, S.S.; Kolbeck, R.C.; Bransome, E.D., Jr., Letter: Ureterosigmoidostomy and cancer: New observations. Annals of internal medicine 1975, 83, 833. Harguindey, S., Hydrogen ion dynamics and cancer: An appraisal. Medical and pediatric oncology 1982, 10, 217-236. Lagadic-Gossmann, D.; Hardonniere, K.; Mograbi, B.; Sergent, O.; Huc, L., Disturbances in H(+) dynamics during environmental carcinogenesis. Biochimie 2019, 163, 171-183.

On the other hand, many sentences have been corrected and shortened in the present format of the MS, as suggested by Referees 2 and 3, and the entire text improved to the best of our capacities. We have also tried our best to avoid redundancies and correct any possible over wording, as referee No 3 has suggested.

Finally, the entire text has been read by a researcher well known in the field of cancer metabolism, having English as his native language. In this vein, either Professor Devesa, the senior author of this publication, or myself, have published more than 250 publications in English for more than 30 years, many times single handedly, and rarely had any negative criticisms regarding our written scientific English.

Unfortunately, any further comments by referee No. 3 are too general and non-specific, so we have not been able to follow his advice and indications more selectively.

Finally, we wish to thank the three reviewers for their work in helping us to improve this manuscript. We honestly think that its quality has been significantly improved after answering one by one, and to the best of our abilities, all the queries, comments and criticisms of each one of the three reviewers. We sincerely hope that the revised manuscript can now be accepted for publication in IJMS.

Salvador Harguindey MD, Ph D.

Round 2

Reviewer 1 Report

There are some improvements in the manuscript. Some minor concerns were suggested to be resolved before the acceptance of this work in IJMS.

My answer for the authors replies ‘First, we believe that the reviewer refers to Table 1, not to Figure 1.’ is No. Except for the minor updates, most parts of the figure are same. Like authors replied ‘Figure 1 has been reformatted as a Table, and the latest information not present in any other previous publication, either of our group or others, inserted in it.’, it is not a new work, therefore, authors should mention the ‘*Modified and updated from ref [13].’ as same way as authors mentioned in the Table 1 legend.

Figure 1 of the submitted manuscript, red boxes are added by reviewer

Please see the following comparison between Table 1 of the submitted manuscript and Table 2 of Ref. 13. There are no differences on the table except the addition of BRCA1 and BRCA2, even the order of listed information is same. Even though the authors added the BRCA family in Table 1, it was not described in the main text. Please explain the relationship between genetic instability/mutation of BRCA family with NHE activity in the main text to support the authors’ claim.

Author Response

Please, see the attachement
